# Prostate Cancer Morphologies: Cribriform Pattern and Intraductal Carcinoma Relations to Adverse Pathological and Clinical Outcomes—Systematic Review and Meta-Analysis

**DOI:** 10.3390/cancers15051372

**Published:** 2023-02-21

**Authors:** Rafał Osiecki, Mieszko Kozikowski, Beata Sarecka-Hujar, Michał Pyzlak, Jakub Dobruch

**Affiliations:** 1Department of Urology, Independent Public Hospital them. prof. W. Orłowskiego, Medical Centre of Postgraduate Education, 00-401 Warsaw, Poland; 2Polish Center of Advanced Urology, Department of Urology, St. Anne’s Hospital EMC, 05-500 Piaseczno, Poland; 3QUADIA MRI Centre, 05-500 Piaseczno, Poland; 4Department of Basic Biomedical Science, Faculty of Pharmaceutical Sciences, Medical University of Silesia in Katowice, 41-200 Sosnowiec, Poland; 5Department of Pathology, Institute of Mother and Child, 01-211 Warsaw, Poland; 6Center for Diagnostic Pathology, 01-496 Warsaw, Poland

**Keywords:** prostate cancer, cribriform pattern, intraductal carcinoma, meta-analysis, radical prostatectomy

## Abstract

**Simple Summary:**

Prostate cancer is one of the most common male cancers. A more accurate disease assessment is needed to better stratify patients’ risks and guide treatment decisions. It has already been studied that some prostate cancer submorphologies are associated with worse outcomes. We performed a systematic review and meta-analysis on the impact of distinct prostate cancer morphologies: the cribriform pattern and intraductal carcinoma on adverse pathological and clinical outcomes after radical prostatectomy. Our results showed that the cribriform pattern together with intraductal carcinoma are negative prognostic factors associated with both adverse clinical and pathological outcomes in the radical prostatectomy cohort, and the presence of those patterns should be implemented in the surgical planning and postoperative treatment guidance.

**Abstract:**

The present study aimed to assess the association between the cribriform pattern (CP)/intraductal carcinoma (IDC) and the adverse pathological and clinical outcomes in the radical prostatectomy (RP) cohort. A systematic search was performed according to the Preferred Reporting Items for Systematic Review and Meta-Analysis statement (PRISMA). The protocol from this review was registered on the PROSPERO platform. We searched PubMed^®^, the Cochrane Library and EM-BASE^®^ up to the 30th of April 2022. The outcomes of interest were the extraprostatic extension (EPE), seminal vesicle invasion (SVI), lymph node metastasis (LNS met), risk of biochemical recurrence (BCR), distant metastasis (MET) and disease-specific death (DSD). As a result, we identified 16 studies with 164 296 patients. A total of 13 studies containing 3254 RP patients were eligible for the meta-analysis. The CP/IDC was associated with adverse outcomes, including EPE (pooled OR = 2.55, 95%CI 1.23–5.26), SVI (pooled OR = 4.27, 95%CI 1.90–9.64), LNs met (pooled OR = 6.47, 95%CI 3.76–11.14), BCR (pooled OR = 5.09, 95%CI 2.23–11.62) and MET/DSD (pooled OR = 9.84, 95%CI 2.75–35.20, *p* < 0.001). In conclusion, the CP/IDC belong to highly malignant prostate cancer patterns which have a negative impact on both the pathological and clinical outcomes. The presence of the CP/IDC should be included in the surgical planning and postoperative treatment guidance.

## 1. Introduction

Prostate cancer (PCa) is the second most common male malignancy worldwide [1]. The introduction of the prostate-specific antigen (PSA) at the end of the 1980s has revolutionized both the diagnosis and the management of the disease. The subsequent years brought a constant decline in prostate cancer-specific mortality in the US [2]. At the same time, numbers of indolent PCa soared and resembled overdiagnosis. It was followed by overtreatment. The mainstay therapy for localized PCa remains either a radical prostatectomy (RP) or radiotherapy. Unfortunately, the two modalities are widely known to be associated with significant morbidity. Therefore, the protocols of active surveillance (AS) aiming at postponing the treatment were eagerly implemented across the world [3]. AS oncological safety has been shown in many studies, and it has become the pivotal therapy of low-risk disease. The success has encouraged a number of investigators to explore the role of AS in intermediate-risk PCa and expand the indications above the lowest risk category. To avoid harm, potential predictive markers of the AS outcome in this specific group of PCa patients are currently under scrutiny. The major advantage of RP is the postoperative histopathological assessment that provides the highest possible accuracy of the staging and grading of PCa. A detailed report embraces valuable information regarding a patient’s prognosis which aids in postoperative clinical decision making. The pathological features in the RP specimen that have a confirmed impact on the treatment outcome include the cancer grade and stage, positive surgical margins (PSM), and lymph node metastasis (LNs met) [4,5,6]. In addition to the prognosticators, there are other emerging pathological features that appear to compromise the oncological outcomes. The 2014 International Society of Urological Pathology Consensus Conference on the Gleason Grading of Prostatic Carcinoma stated that the cribriform pattern (CP) should be considered part of the spectrum of the Gleason Grade 4 pattern along with glomeruloid, fused and poorly formed glands [7]. Previous studies indicate its association with the extraprostatic extension (EPE), PSM, biochemical recurrence (BCR), distant metastasis (MET) and disease-specific death (DSD).

Another pathological PCa entity that may have a dismal prognosis is intraductal carcinoma (IDC). In 2006, Guo and Epstein published the most commonly used morphological description of IDC [8]. The WHO Classification of Prostatic Tumors in 2016, for the first time, recognized IDC as a new entity and provided a detailed histopathological description of it. [9]. The incidence of IDC in an RP specimen ranges from 20–40%, depending on the tumor grade and stage [10,11]. The studies conducted so far associate the presence of IDC with the Gleason pattern (GP) 4 and 5, a more advanced tumor stage, an increased risk of BCR and shorter MET-free survival [12,13,14,15,16,17]. From a pathological point of view, IDC and the CP show architectural similarities and, as a result, can sometimes be misinterpreted. It has been confirmed that they might coexist in the same tumor [18,19]. Therefore, in equivocal cases, additional immunohistochemical staining for basal cells is recommended. Despite that, sometimes it is impossible to distinguish between the two patterns.

In the present study, we aimed to investigate the correlation of both the CP and IDC with pathological as well as clinical outcomes by performing a systematic review and meta-analysis of the published data.

## 2. Materials and Methods

### 2.1. Pathological Definition

In the present study, only articles with pathological assessment based on the 2005 or 2014 ISUP guidelines were analyzed. The 2005 ISUP guidelines recommended CP to be graded as GP4 but allowed some cribriform glands to be included as GP3 [20]. Among various changes that the 2014 ISUP conference brought to the pathological assessment of the RP specimen, one of the most important was the recommendation to include all cribriform glands as one of the GP4 spectrum morphologies [7]. However, until 2021, there was no uniform definition of CP. Nonetheless, a study by Kweldam et al. [21] proved that CP had good interobserver reproducibility among pathologists, unlike other Gleason 4 patterns, such as ill-formed or fused glands. To further improve the pathological assessment of CP, in 2021, ISUP provided consensus definition of CP. Currently, CP is defined as a confluent sheet of contiguous malignant epithelial cells with multiple glandular lumina that are easily visible at low power. There should be no intervening stroma or mucin separating individual or fused glandular structures [22].

In 2016, WHO recognized IDC as a new entity, not included in the Gleason classification, and defined it as an intra-acinar and/or intraductal neoplastic epithelial proliferation that has some features of high-grade prostatic intraepithelial neoplasia (HGPIN), exhibiting much greater architectural and/or cytological atypia, typically associated with high-grade and/or high-stage PCa [9]. The 2022 EAU Prostate Cancer Guidelines require reporting of both patterns in prostate biopsy and RP specimens. These guidelines, however, do not specify whether immunohistochemical staining is demanded to distinguish those entities [3]. Even the absence of basal cells cannot be considered pathognomonic for invasive CP as they may not be visible on a particular slide [18]. CP and IDC can morphologically mimic each other and even belong to a pathological and biological continuum [23,24]. Kryvenko et al. [25] found that the frequency of IDC was related to the proportion of CP. Kweldam et al. [19] reported that CP and IDC tend to coexist in both biopsy and RP specimens. Therefore, we decided to assess both entities CP and IDC together.

### 2.2. Evidence Acquisition

This systematic review and meta-analysis were conducted in accordance with the Preferred Reporting Items for Systematic Reviews and Meta-analyses (PRISMA) guidelines [26]. The protocol was registered in PROSPERO (CRD42020183346). The research question for this systematic review was established in the PICO (population, intervention, comparison and outcomes) framework, which was as follows: What is the prognostic value of PCa patterns: CP/IDC for clinical and pathological outcomes in men who underwent radical prostatectomy?

### 2.3. Literature Search

A thorough literature search was conducted, PubMed^®^, the Cochrane Library and EMBASE^®^ databases, with the following keywords: (“prostatic neoplasms” OR “prostate cancer” OR “pca” OR “prostate” OR “prostatic”) AND (“intraductal” OR “intraductal cancer” OR “intraductal carcinoma” OR “IDC” OR “cribriform” OR “cribriforming”). The search was restricted to English language publications and included every article fulfilling the inclusion criteria from the inception of the databases until 30 April 2022. The Mendeley Desktop version 1.19.4 citation manager was used to store records and remove duplicates. For completeness, references to incorporated articles and review articles were also extensively searched.

Two investigators (R.O. and M.K.) independently screened and assessed article eligibility based on their titles and abstracts, adopting the PICO approach. Any disagreements were resolved by discussion with a senior author (J.D.). Eligible were original studies with a cohort of patients with PCa, whose initial treatment modality was RP (surgical approach was not considered). Additional inclusion criteria were a comparison of adverse pathological/clinical outcomes between PCa with and without CP/IDC.

Both investigators extracted data from full-text publications and cross-checked values. In the case of multiple reports of the same cohort, the most complete data were selected. We excluded articles not in English, reviews, editorials, case reports and animal or in vitro studies. Additionally, references where CP/IDC was evaluated in a prostate biopsy, or where data were not stratified, or the authors were not able to extract data by the presence of CP/IDC, or the record did not provide data on prognosis/adverse outcome were also excluded. The selection process is shown in Figure 1.

### 2.4. Data Extraction

Data extracted from eligible studies and study characteristics are presented in Appendix A. Despite direct e-mail contact with authors, it was not possible to obtain more precise data from 3 studies [17,27,28]. The adverse outcomes reported in the analyzed studies included EPE, SVI, LNs met, BCR, local recurrence (LR), clinical recurrence (CR), MET, DSD, cancer-specific survival (CSS), overall survival (OS). The PSM as an outcome measure was excluded from this systematic review and meta-analysis because it does not necessarily confer non-organ-confined disease and may result from surgical flaws, postsurgical tissue handling or misinterpretation. Quantitative analysis was performed wherever data for calculations were available.

### 2.5. Assessment of Study Quality

The assessment of the risk of bias in the studies included in the meta-analysis was conducted with the Newcastle–Ottawa Quality Assessment Scale (NOS), which was developed to assess the risk of bias in observational studies [29]. We decided to dismiss the response rate in the exposure domain in case-control studies as this subdomain was not applicable to this article. Each study could receive a maximum of 8 stars for case-control studies and 9 stars for cohort studies. Two investigators (R.O. and M.K.) independently evaluated each article, any disagreement was discussed with and resolved by the third senior investigator (J.D.). Observational comparative studies with scores of <4, 4–6 and >6 were considered, having a high, intermediate and low risk of bias, respectively. According to the NOS, 11 studies were considered high quality and 2 were evaluated as intermediate quality (Appendix A).

### 2.6. Statistical Analysis

Statistical analyses were performed with the use of Review Manager software (RevMan version 5.4 Cochrane, UK) and StatsDirect 3 link software (version 3.3.5; StatsDirect Ltd. Wirral, UK). The strength of associations between CP/IDC and pathological as well as clinical outcomes were analyzed by a calculation of the pooled odds ratios (OR) with 95% confidence interval (CI). The degree of heterogeneity between included studies was estimated with the I2 test which describes the proportion of variance (from 0% to 100%). The random effects method (DerSimonian–Laird; REM) was used to calculate the pooled OR with the 95% CI. The obtained results were summarized in tables as well as illustrated using forest plots. Potential publication bias was analyzed with Egger’s regression and Begg’s rank correlation tests for all comparisons. In addition, sensitivity analyses were made by sequential exclusion of each study to evaluate the stability and in turn reliability of the results.

## 3. Evidence Synthesis

### Characteristics of Included Studies

The article selection process was conducted according to the PRISMA statement, and a total of 1798 references were identified. Of 16 studies included in the systematic review, 13 articles provided sufficient data for a quantitative evaluation. The studies included in this systematic review were published between 2011 and 2020.

The systematic review encompassed 164,296 patients. The sample sizes varied significantly among the studies, ranging from 28 to 159,777 participants. It is noteworthy that cohorts from two studies partially overlapped, but the endpoint measures were different; therefore, both studies were included and assessed separately [30,31]. The statistical analysis was based on 3254 patients; of those, 1405 had the CP/IDC morphology (43%). All the included studies were retrospective (16/16). Most of them were case-control studies (14/16), two of which were designed as paired and nested paired case-control studies [25,32]. Additionally, there was one population-based study and one cohort study [13,33].

Eight studies focused on the CP [28,30,31,32,33,34,35,36], whereas five evaluated IDC [10,13,14,17,25]. Three studies allowed for a combined assessment of both patterns [27,37,38]. The pathological evaluation of resected specimens was conducted according to the ISUP 2005 criteria in six studies [25,27,30,31,32,39]. The ISUP 2014 Consensus Conference Working Group criteria were adopted by the authors of six studies [14,17,28,34,37,38]. In the study by Kweldam et al. [36], the adopted histopathological criteria differed depending on the time of the RP. Two articles did not include detailed information regarding a pathological assessment [10,33]. In the study by Dinerman et al. [13], the data on the histopathological evaluation were not provided.

Two studies investigated the large (LC)/expansile CP (EC) [27,33]. According to the 2021 WHO recommendation, the CP should be reported without subtypes. Therefore, we considered the LC as the CP. Luo et al. [28] focused on invasive cribriform lesions (ICLs), which we also included as the CP. In the studies which focused on IDC, this pattern was defined using the Guo and Epstein criteria in four studies [14,25,27,38]. Kato et al. [17] adopted the McNeal and Yemoto definition of IDC. Miyai et al. [10] defined IDC as one or both of the following patterns: (1) solid or dense cribriform (less than 50% lumen in a duct) intraductal lesions or (2) loose cribriform (50% and more than 50% lumen in a duct) or micropapillary intraductal lesions with prominent nuclear pleomorphism (nuclear size greater than 6× normal) and/or non-focal comedonecrosis. In the study by Hollemans et al. [37], IDC was identified if the cribriform structures were clearly continuous with the pre-existing glands lined by normal basal epithelium or containing corpora amylacea. The presence or absence of the CP/IDC in the evaluated specimen was reassessed by at least two specialized genitourinary pathologists in 62.5% of the studies (10/16). Luo et al. [28] stated only that the slides were reviewed for the purpose of the study. In two other studies, the diagnosis of the CP/IDC was extracted from pathology reports [10,33]. In the remaining articles, this information was either not disclosed or there was a single reviewer [17,30]. Dinerman et al. [13] extracted data on the presence of IDC from the SEER database. In equivocal cases, the authors of nine papers used immunohistochemical staining (IHC) to distinguish IDC from the CP or HGPIN [10,14,27,28,30,31,36,37,38], whereas the authors of three papers relied solely on the cancer cell morphology [32,33,34]. In four articles, the information regarding the use of immunohistochemical staining was not disclosed [13,17,25,39].

The median follow-up period in the studies that assessed the BCR varied significantly from 360.5 days to 130.6 months. For the MET or DSD, the median follow-up duration also differed across the studies and ranged from 20 to 120 months. The BCR definitions varied among the studies with a serum PSA level ≥ 0.2 ng/mL being the most commonly used threshold (6/9) [10,32,33,34,37,38]. In two studies, the BCR was defined as a PSA ≥ 0.1 ng/mL (2/9) [28,31], and in one case, the definition differed depending on the period of the sample and data collection [39]. Although Kweldam et al. [36] in their study investigated the BCR, the presented data were insufficient for a quantitative analysis. The LR was investigated by two authors [14,28]. Both authors assessed either the CP or IDC in RP specimens. Trinh et al. [14] defined the LR as a peri-prostatic involvement after surgery as confirmed by imaging, biopsy and/or suspect results from a digital rectal examination (DRE), followed by exclusive radiotherapy to the prostatic bed leading to a serum PSA-level reduction without concurrent hormone therapy. Luo et al. [28] did not elaborate on the LR definition.

MET was also diagnosed differently. In the studies by Kweldam et al. and Trinh et al., the diagnosis of MET required either radiological or pathological confirmation [14,36]. Dong et al. relied only on imaging and Hollemans et al. either on a biopsy or multidisciplinary consensus [37,39]. In one study, the diagnostic criteria for MET were not defined [28]. None of the authors included information regarding the imaging modality used for the confirmation of metastatic spread.

Two studies provided additional value for this systematic review. Luo et al. [28] conducted a systematic review and meta-analysis on the impact of the CP in prostate biopsy/RP specimens on adverse outcomes. The studies included in the analysis by Luo and the current systematic review had some overlap, but the addition of articles evaluating IDC in the RP cohort extended the scope of the current study (see Discussion). Greenland et al. [33] assessed the DECIPHER score in case it was either positive for the CP or the glomerulation pattern. The DECIPHER assay was developed to predict the risk of metastases within 5 years after a prostatectomy and measures the expression levels of the RNA of 22 genes in the radical prostatectomy specimen. A DECIPHER score < 0.45 was interpreted as low risk, 0.45–0.6 corresponded to average risk and > 0.6 was assessed as high risk. The authors of that study emphasize that the assay was performed at the discretion of the treating physician, and patients with negative pathological risk factors were more likely to undergo genetic testing. Nonetheless, CP-positive patients were characterized by higher mean DECIPHER scores than patients with glomerulation patterns (0.61 vs. 0.47; *p* = 0.02; SD = 0.18, 0.17).

Studies by Dinerman et al., Kato et al. and Trudel et al. were qualitatively assessed in the systematic review, but because of either different study concepts or a lack of detailed statistical data, they were disqualified from the meta-analysis (see Discussion) [13,17,27].

## 4. Results

### 4.1. Correlation between EPE and CP/IDC

The correlation between the EPE and CP/IDC was analyzed in eight studies, with a total number of 903 patients with CP/IDC and 1285 without CP/IDC. Significant heterogeneity was observed between the studies (I2 89%). The random effect models analysis revealed that the EPE was significantly more common in patients with than without CP/IDC (41.3% vs. 17.7%, respectively: pooled OR = 2.55 95%CI 1.23–5.26, *p* = 0.01) (Figure 2).

### 4.2. Correlation between SVI and CP/IDC

The authors of seven studies presented information regarding the relationship between SVI and the CP/IDC. In this meta-analysis, there were 788 patients with CP/IDC and 1215 without CP/IDC, respectively. SVI was significantly more common in patients with CP/IDC than without (19.9% vs. 4.9%), respectively, which resulted in a pooled OR = 4.27 (95%CI 1.90–9.64, *p* < 0.001) (Figure 3).

### 4.3. Correlation between LNs met and CP/IDC

Seven studies provided data for the analysis of the LNs met in correlation with the CP/ICD, with a total of 639 patients with CP/IDC and 1217 patients without CP/IDC. The LNs met were significantly more prevalent in patients positive for CP/IDC (21.6% vs. 5.8%), with a pooled OR = 6.47 (95%CI 3.76–11.14, *p* < 0.001) (Figure 4).

### 4.4. Correlation between BCR and CP/IDC

Nine studies analyzed the association between the BCR and CP/IDC. The total number of patients with CP/IDC was 1023 and there were 1498 cases without CP/IDC. The random effect models analysis showed that the BCR was significantly more frequent in individuals positive for CP/IDC (32.2% vs. 8.0%), with a pooled OR = 5.09 (95%CI 2.23–11.62, *p* < 0.001) (Figure 5).

### 4.5. Correlation between MET/DSD and CP/IDC

MET/DSD as an adverse outcome was analyzed in five studies, with a total number of 581 positive for CP/IDC and 411 without CP/IDC. The random effect models analysis revealed that MET/DSD was also significantly more common in patients with CP/IDC (23.4% vs. 2.9%), which resulted in a pooled OR = 9.84 (95%CI 2.75–35.20, *p* < 0.001) (Figure 6).

In the sensitivity analysis, no changes in the OR value were demonstrated in all the performed comparisons after excluding the subsequent studies. Therefore, the above-described analyses were considered stable and reliable.

### 4.6. Publication Bias

The publication bias tests indicated that there is no publication bias in the studies on the relationship between the EPE, SVI, LNs met, BCR, MET/DSD and CP/IDC. The exact results of Egger’s regression and Begg’s rank correlation tests, i.e., the intercept with a 95%CI and Kendall’s tau are shown in Appendix A. In addition, the publication bias was evaluated visually with the inspection of funnel plots, which were eventually found to be symmetric (Figure 7).

## 5. Discussion

We performed this systematic review and meta-analysis to investigate the impact of the CP/IDC on the adverse pathological and clinical outcomes after an RP. To the best of our knowledge, this is the first study to systematically and quantitively assess the prognostic factors of CP/IDC in the RP cohort. The results of this study indicate a relationship between the CP/IDC and unfavorable prognostic factors, such as the advanced stage of the disease, BCR, MET and DSD.

The aim of a radical prostatectomy is to remove the whole prostatic gland, seminal vesicles and distal part of the vas deferens with the adjacent periprostatic tissue in select cases to ensure the best oncological outcome. Proper surgical planning includes neurovascular bundle (NVB)-sparing techniques in order to restore continence and erectile function after surgery [40]. To date, no recommendations have been released regarding the extent of the surgery in cases with CP/IDC morphology. The results of the meta-analysis clearly indicate a close relationship between the EPE/SVI and CP/IDC (OR = 2.55 and OR = 4.27, respectively). Although we decided to exclude the PSM from this systematic review and meta-analysis, as explained earlier, the PSM has an added value with regard to an oncological prognosis. The articles included in the systematic review reported inconclusive results on the association between the CP/IDC and PSM. According to Sarbay et al. [30], GG1 cases with a CP have been found to have a much higher incidence of PSM (23.1% vs. 5.1%, *p* < 0.011), but no such correlation was observed in patients with GG2 and 3 PCa (*p* < 0.331). Dong et al. [39] and Kweldam et al. [36] also did not report an association between the CP and PSM. However, in two other studies, PSMs were more commonly found after an RP in the presence of the CP [34,37]. PSMs in the presence of IDC were also investigated. In a retrospective population-based analysis, the PSM was found to be more commonly encountered in the presence of IDC (25.6% vs. 19.5%, *p* = 0.02) [13]. According to Trinh et al. [14], there was not any significant correlation between IDC and the PSM. Contrary to our results, Flammia et al. [41] found no correlation between the CP and the local disease stage, the SM in GG 2–5 RP patients. The EAU recommends offering NVB-sparing surgery to patients with a low risk of EPE, which is, among other factors, based on the GG [3]. GP 4 itself does not exclude the NVB-sparing approach, but those guidelines do not take into consideration the GP4 submorphologies or IDC. The results of our meta-analysis indicate a higher prevalence of both the EPE and SVI in CP/IDC cases. Therefore, in the presence of the CP/IDC, intrafascial NVB preservation may pose a risk of a PSM.

We also identified the CP/IDC in the radical RP as being associated with LNs met. These results are consistent with our previous study, which investigated the clinical importance of the CP in a prostate biopsy [42]. Nodal metastases, especially in greater numbers, are a known negative prognostic factor associated with a worse BCR-free survival, MET-free and OS [43,44,45,46,47]. Unfortunately, the included studies did not provide detailed data on the extent of the LN dissection (LND) and the number of dissected LNs, which made it impossible to draw meaningful conclusions regarding the true significance of the LNs met in the presence of the CP/IDC. At the same time, especially extending the LND during an RP provides more tissue for a histopathological analysis. Detailed information on the disease stage may help guide the adjuvant treatment. The extent of the LND and the role of adjuvant therapy in the presence of the CP/IDC and other postprostatectomy negative prognostic factors have yet not been fully understood.

The results of our meta-analysis clearly indicate the negative impact of the CP/IDC on the long-term oncological outcomes such as the BCR, MET and DSD and are consistent with the results presented in the literature [12,35,36]. Moreover, a meta-analysis by Luo et al. [28] also revealed the negative impact of the CP on the BCR, MET and DSD, which is understandable as the articles included by Luo et al. partially overlapped with those evaluated in our study. However, a study by Flammia et al. [41] showed that no association between the CP and BCR was found in the GG 2–5 subjected to an RP with a median follow-up of 22 months. A comprehensive systematic review and meta-analysis conducted by Miura et al. [12] on the clinical importance of IDC in localized and advanced PCa showed that in the case of localized PCa, the presence of IDC in either a biopsy or radical prostatectomy specimen was associated with worse BCR-free survival (pooled HR 2.09) and CSS ((pooled HR = 2.93). Additionally, there was a correlation between IDC and shorter OS in advanced PCa (pooled HR = 2.93). Dinerman et al. [13] found a 3-fold higher risk for DSD in the presence of IDC but reported no association between IDC and OS. BCR-free survival in more than 1000 RP patients was significantly worse in any ISUP with IDC than without in a study by Kato et al. Moreover, GG2 men with IDC had a prognosis similar to GG4 or GG5 patients without this morphology [17]. Trudel et al. [27], who investigated the BCR-free rate, concluded that in the presence of LC/IDC, the BCR occurred more commonly, and again, any GG without LC/IDC had a better prognosis. In a subgroup analysis, Kaplan–Mayer curves for the BCR-free survival were similar in men with LC and in men with LC/IDC. IDC as an isolated morphology was, on the other hand, associated with worse outcomes. Nonetheless, the authors highlighted that isolated IDC was present in only 10 men and this cohort was too small to draw meaningful conclusions [27]. Trinh et al. [48] investigated the effect of adjuvant radiotherapy (ART) on the BCR in patients with IDC and found that the worst BCR-free survival was in men positive for IDC, who did not undergo ART (log-rank *p* = 0.023).

AS oncological safety has been shown in many studies. As a result, it has become the pivotal therapy of low-risk disease, acknowledged by many international and national urological associations [3,49]. The success has encouraged a number of investigators to explore the role of AS in intermediate-risk PCa and expand the indications above the lowest risk category. Decisions regarding AS are made based on the results of a prostate biopsy. This systematic review and meta-analysis refers to the RP histopathological assessment which is more detailed and provides the most accurate evaluation of the disease stage and grade. In our study, CP/IDC emerge as negative prognosticators in terms of both pathological and clinical outcomes after an RP. Considering our results, the negative impact of the CP/IDC should be discussed with every patient qualified for AS. This statement is supported by international urological associations. The current EAU guidelines allow AS in highly selected GG2 cases (<10% GP4, PSA< 10 ng/mL, cT2a); the identification of CP/IDC in a biopsy sample is considered as an absolute contraindication for this type of deferred treatment [3]. Similarly, the PCa guidelines provided by the America Urological Association (AUA) recognize CP/IDC in intermediate-risk patients as unfavorable characteristics and therefore do not recommend AS in those cases [49]

Several factors could not be included in the meta-analysis due to insufficient data. In the systematic review by Luo et al. [28], we found that the risk of LR is more than twice as high in patients with the CP (OR 2.32). Trinh et al. [14] showed that IDC was linked to distant metastasis (mainly bone metastases) as the initial site of the CR on both univariate and multivariate analyses (OR = 6.27), but surprisingly, the time to the CR and CSS was not significantly different between men with and without IDC. A subgroup analysis revealed that radiotherapy (adjuvant or salvage) was superior to pre-RP ADT in respect to the CSS in men with IDC (170 months, 95%CI (124–215) vs. 159 months, 95%CI (122–196)). That was one of the first studies to highlight the detailed evaluation of a pathological specimen and IDC detection in terms of post-RP management. A single study included in this meta-analysis evaluating the impact of the CP on the OS found the OS in GS7 (includes both GG 2 and 3) men with CP to be shorter (log-rank *p* = 0.001) [36].

The malignant potential and basis for the joint assessment of CP/IDC are reflected in the genomic and epigenetic alternation observed in these histopathological patterns. Elevated SChLAP1 expression and well as genomic instability were found in PCa with CP/IDC [50]. Mehra et al. [51,52] indicated that increased SChLAP1 expression was associated with a higher risk of BCR, DSD and MET. Based on the whole slide images of the TGGA database and dataset from CPCGN, Böttcher et al. [53] found that IDC and the CP were associated with an increased genomic instability that affected specific regions: the chromosomal deletions of 3p13, 6q15, 8p21–23, 10q23, 13q14 16q21–24 and 18q21–23, and the amplification of chromosome 8q24. Those regions are known to be involved in aggressive PCa, for example, the BRCA 2 gene is located on the long arm of chromosome 13. In CP/IDC-positive cases, genomic alternations also affect the epigenetic profile. Olkhow-Mitsel et al. [54] reported that CP/IDC cases were characterized by DNA hypermethylation in the APC, RASSF1 and TBX15 genes. Hypermethylation in those genes has already been linked to more aggressive PCa [55]. The authors also suggested the utilization of methylation profiles in those genes as a marker of CP/IDC [54]. In a study by Taylor et al. [56], who investigated the correlation between the CP and IDC with the DECIPHER genomic classifier, found that the predominance of CP in RP specimens significantly increased the DECIPHER score. IDC was non-significantly associated with an increased score (OR 1.92 *p* = 0.24). On the other hand, the presence of CP/IDC did not reach statistical significance in terms of high-risk categorization (OR = 4.53, *p* = 0.08), which could be due to the small sample size (11 men) [56]. In a study by Risbridger et al. [57], which investigated patients-derived xenografts, BRCA 2 germline mutations were more common in IDC-positive men with localized PCa. In contrast to those results, Lozano et al. [58] found no association between the germline BRCA 2 mutations and CP/IDC morphology. In the same study, however, the bi-allelic BRCA2 loss in primary PCa was significantly correlated with the CP and IDC. The authors, therefore, advised against the routine germline BRCA2 testing in the presence of CP/IDC but felt that further research was needed to investigate the relevance of CP/IDC as a marker of the bi-allelic BRCA2 loss in primary PCa.

The present study has several limitations. The key one is the different diagnostic criteria applied by authors for the identification of CP/IDC. Various definitions of IDC used by authors and high interobserver variation in the identification of IDC might have contributed to its under- or overdiagnosis in the selected articles. Additionally, the histopathological evaluation of the RP specimens was conducted according to either the ISUP 2005 or 2014 diagnostic criteria. Therefore, the true prevalence of CP/IDC might differ from what was reported. Moreover, the definitions of BCR and MET varied across the studies, which in turn might have had an influence on the presented results. Our conclusion to exclude patients with CP/IDC from the AS is based on studies in which the RP specimen was assessed and not the prostate biopsy. Nonetheless, this attitude is supported by international urological associations. Although a statistical analysis revealed that CP/IDC compromise the long-term oncological outcome in the RP cohort, an analysis of the impact of radiotherapy on patients with CP/IDC might contribute meaningful insight into the prognosis in the setting of different radical treatment modalities.

## 6. Conclusions

The CP and IDC should be considered as highly aggressive histopathological morphologies of the prostatic adenocarcinoma, which have a negative impact on both the pathological and oncological outcomes. The presence of CP/IDC should be taken into account in the planning of surgery and proper postoperative counselling. In intermediate-risk patients with either of the two entities, a detailed explanation of their higher malignant potential is required and AS should be discouraged, in line with international urological guidelines.

## Figures and Tables

**Figure 1 cancers-15-01372-f001:**
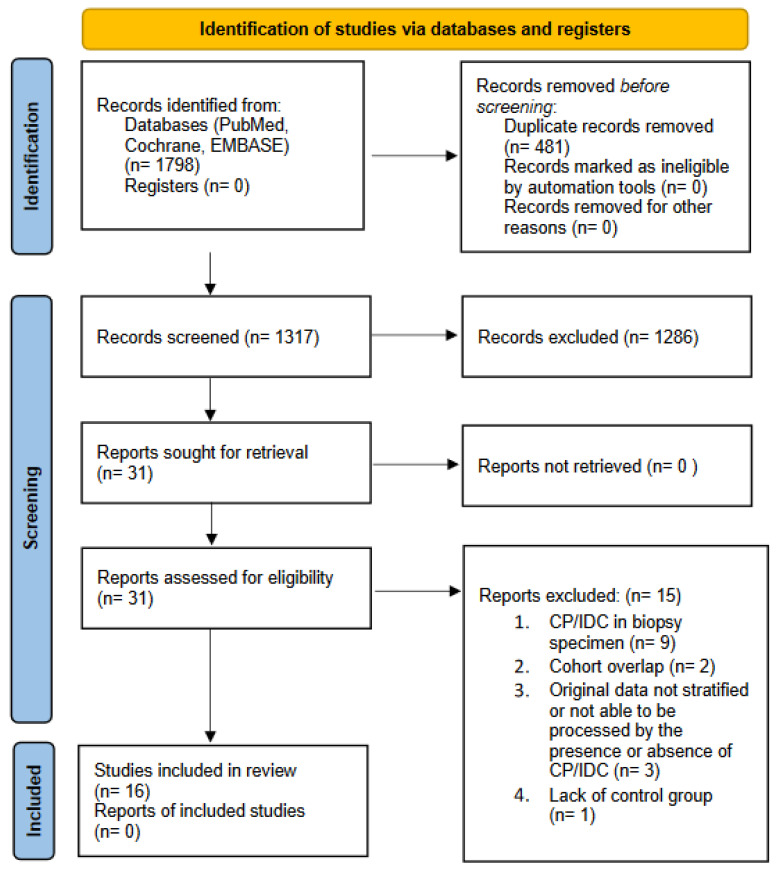
Literature search, screening and selection for inclusion in systematic review.; CP = cribriform pattern; IDC = intraductal carcinoma.

**Figure 2 cancers-15-01372-f002:**
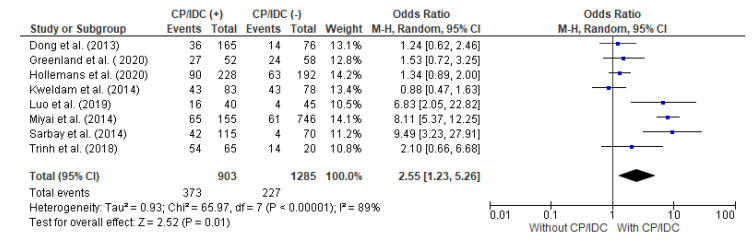
Forest plot of odds ratios (ORs) in the studies analyzing the relationship between EPE and CP/IDC [10,14,28,30,33,36,37,39]. M.-H. = Mantel–Haenszel; CI = confidence interval; I^2^ = heterogeneity; df = degrees of freedom; EPE = extraprostatic extension; CP = cribriform pattern; IDC = intraductal carcinoma. Each blue square in Figure 2 represents an effect size of a study and the area of the square represents the magnitude of a related study in the effect size. The lines on either side of the squares indicate the lower and upper limits in a 95%CI of the calculated effect sizes. The black rhombus at the bottom of the plot shows the calculated overall effect size.

**Figure 3 cancers-15-01372-f003:**
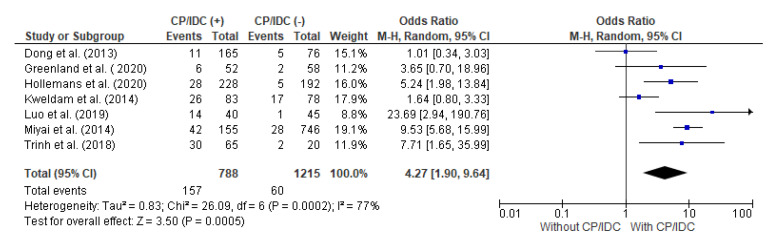
Forest plot of odds ratios (ORs) in the studies analyzing the relation between SVI and CP/IDC [10,14,28,33,36,37,39]. M.-H. = Mantel–Haenszel; CI = confidence interval; I^2^ = heterogeneity; df = degrees of freedom; SVI = seminal vesicle invasion; CP = cribriform pattern; IDC = intraductal carcinoma. Each blue square in Figure 3 represents an effect size of a study and the area of the square represents the magnitude of a related study in the effect size. The lines on either side of the squares indicate the lower and upper limits in a 95%CI of the calculated effect sizes. The black rhombus at the bottom of the plot shows the calculated overall effect size.

**Figure 4 cancers-15-01372-f004:**
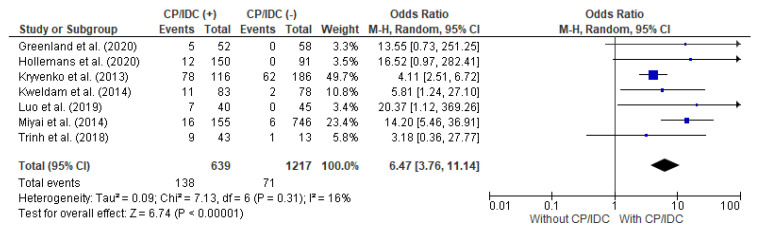
Forest plot of odds ratios (ORs) in the studies analyzing the relation between LNs met and CP/ICD [10,14,25,28,33,36,37]. M.-H. = Mantel–Haenszel; CI = confidence interval; I^2^ = heterogeneity; df = degrees of freedom; LNs met = lymph node metastasis; CP = cribriform pattern; IDC = intraductal carcinoma. Each blue square in Figure 4 represents an effect size of a study and the area of the square represents the magnitude of a related study in the effect size. The lines on either side of the squares indicate the lower and upper limits in a 95%CI of the calculated effect sizes. The black rhombus at the bottom of the plot shows the calculated overall effect size.

**Figure 5 cancers-15-01372-f005:**
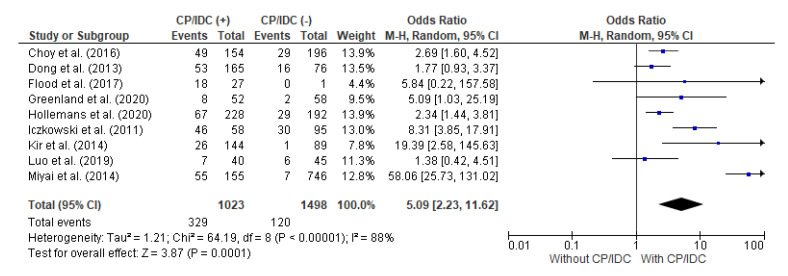
Forest plot of odds ratios (ORs) in the studies analyzing the relation between BCR and CP/ICD [10,28,31,32,33,34,37,38,39]. M.-H. = Mantel–Haenszel; CI = confidence interval; I^2^ = heterogeneity; df = degrees of freedom; BCR = biochemical recurrence; CP = cribriform pattern; IDC = intraductal carcinoma. Each blue square in Figure 5 represents an effect size of a study and the area of the square represents the magnitude of a related study in the effect size. The lines on either side of the squares indicate the lower and upper limits in a 95%CI of the calculated effect sizes. The black rhombus at the bottom of the plot shows the calculated overall effect size.

**Figure 6 cancers-15-01372-f006:**
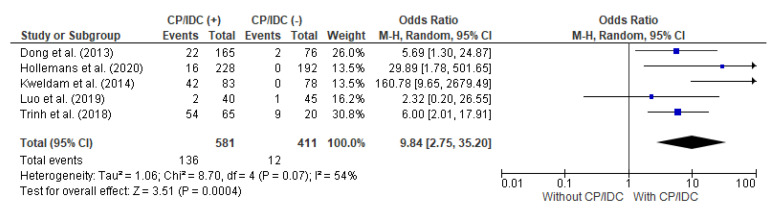
Forest plot of odds ratios (ORs) in the studies analyzing the relation between MET/DSD and CP/IDC [14,28,36,37,39]. M.-H. = Mantel–Haenszel; CI = confidence interval; I^2^ = heterogeneity; df = degrees of freedom; MET = distant metastasis; DSD = disease-specific death; CP = cribriform pattern; IDC = intraductal carcinoma. Each blue square in Figure 6 represents an effect size of a study and the area of the square represents the magnitude of a related study in the effect size. The lines on either side of the squares indicate the lower and upper limits in a 95%CI of the calculated effect sizes. The black rhombus at the bottom of the plot shows the calculated overall effect size.

**Figure 7 cancers-15-01372-f007:**
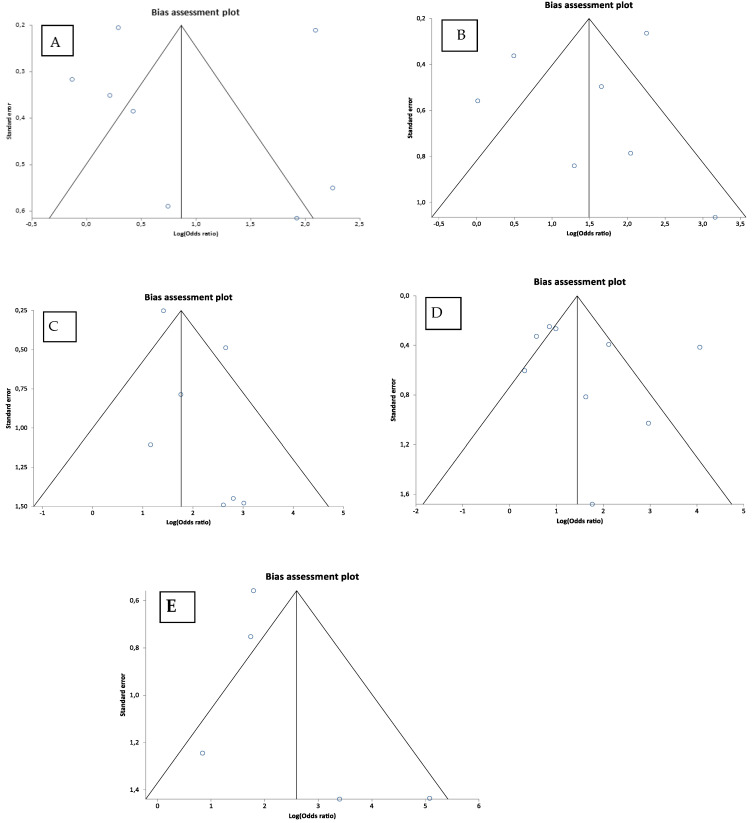
Funnel plots of the correlation between extraprostatic extension (**A**); seminal vesicle invasion (**B**); lymph node metastasis (**C**); biochemical recurrence (**D**); distant metastasis/disease-specific death (**E**) and cribriform pattern/intraductal carcinoma in the studies included. Each circle represents a study included in the analysis.

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
