# Peer review of "Prostate Cancer Morphologies: Cribriform Pattern and Intraductal Carcinoma Relations to Adverse Pathological and Clinical Outcomes—Systematic Review and Meta-Analysis"

_cancers, 2023, doi:10.3390/cancers15051372_

Round 1
Reviewer 1 Report
Very well written paper. Two comments:- in the discussion, it s mention (ligne 442), that one of the limitations, the analysis of the impact of radiotherapy on patients with IDC; Our group from UM, published on the benefit of adjuvant radiotherapy for patients with IDC. See ref
![]() |
Retrospective study on the benefit of adjuvant radiotherapy in men with intraductal carcinoma of prostate - PubMed Men with IDC-P who did not receive ART had the highest BCR rates, and IDC-P had the same impact as ≥1 HRF, which are often used as ART indications. Once validated, ART should be considered in patients with IDC-P. pubmed.ncbi.nlm.nih.gov |
2.the paragraph from ligne 104 to 108 is already mention in the introduction ligne 73-77 Thank you
Reviewer 2 Report
Overall, the management of prostate cancer and clinical outcomes after treatments are significantly influenced by histological factors. Indeed, heterogeneity in histology can be one of the explanations for the variable outcomes. Specifically, cribriform architecture is one of the 4 growth patterns recognized in Gleason pattern 4 (poorly formed, fused glands, glomerulation, and cribriform types) prostate cancer according to the ISUP 2014 classification. Another pathological prostate cancer entity that may negatively affect prognosis is intraductal carcinoma .
Even though the concept of heterogeneity in histology may be associated with worsening of prostate cancer patient prognosis and outcomes, its significance and diagnostic relevance remain controversial.
To date, several studies have investigated the correlationof both cribriform pattern and intraductal with pathological as well as clinical outcomes.
In the present paper, the authors sought to perform a systematic review and meta-analysis on the impact of these two prostate cancer morphologies.
Overall, the manuscript is well written and close attention has been paid to the accuracy of the text.
Title: accurate
Abstract: reflects the report
Introduction: clearly states background and objectives
M&M: study design is clear
Results: the data are well presented. However, a Table summarizing main data from included studies [i.e.. study type (single-centre or multi-centre, prospective or retrospective, etc.), level of evidence (I: High quality randomized trial or prospective study, systematic review of Level I RCTs and Level I studies; II: Lesser quality RCT; prospective comparative study; retrospective study; etc.), numbers of included patients, et.] would be very useful for readers
Discussion: the authors revise related literature
Conclusions: the interpretation is clear
References: I would suggest to check and update the references list. I.e. please see and eventually include the following paper: Flammia S et al. Cribriform pattern does not have a significant impact in Gleason Score ≥7/ISUP Grade ≥2 prostate cancers submitted to radical prostatectomy. Medicine (Baltimore). 2020 Sep 18;99(38):e22156. doi: 10.1097/MD.0000000000022156. PMID: 32957339; PMCID: PMC7505347.
Reviewer 3 Report
This study performed a systematic review and meta-analysis regarding the impact of CP/IDC on adverse pathological and prognostic outcomes. It showed that CP/IDC was significantly associated with EVE, SVI, LN mets, BCR, MET, and DSD.
1) This study focused on CP/IDC in RP specimens and excluded studies that evaluated CP/IDC in biopsy specimens. Thus, I think that the no recommendations regarding AS are not obtained from the results of this study. However, the authors commented AS in patients with CP/IDC in biopsy specimens in the Introduction, Discussion, and Conclusion. These parts should be revised.
2) How do LN dissection and nerve sparing affect BCR, MET, and DSD in patients with CP/IDC? This point should be discussed.
Reviewer 4 Report
This systemic review and Meta-Analysis of prostate cancer revealed that CP/IDC have a negative impact on both pathological and clinical outcomes, and the authors recommended that the presence of CP/IDC should be included in clinical management. My comments are as follow:1. Although the adverse effects of CP/IDC patterns have recently been widely studied, this meta-analysis study further confirmed such effects. I agree on the combination of CP and IDC in the study, mainly due to their similar behavior and the frequent difficulty in distinguishing these two histologically and immunohistochemically.
2. CP/IDC were recommended to be included in the pathology report only in the past few years. The 13 previous studies included in this study were published between 2011-2020. How information of CP/IDC was collected in these 13 studies should be briefly mentioned in the current study, such as retrospective review of the slides of individual cases or based on the description in pathology reports.
3. Glomeruloid pattern was recently excluded from being called cribriform pattern because of its correlation with a better prognosis than CP. The specification of glomeruloid pattern in the 13 studies should be clarified or discussed.
4. PSM (positive surgical margin) is a common indicator of incomplete tumor resection and reporting of PSM is a standard practice. Based on my experience, given that prostate is a very solid organ and most radical prostatectomy are assisted by robot, the surgical flaws, tissue handling, or mis-interpretation are rarely the causes of PSM. Abnormal postoperative PSA levels can be related to residual tumor from incomplete resection or tumor metastasis. Exclusion of PSM in the study could have compromised the conclusion made to the negative effects of CP/IDC on BCR and CR which are established mostly based on persistent or elevated PSA levels.
5. The high interobserver variation in the diagnosis of IDC, debatable nature of the IDC (retrograde spread from invasive PC vs precursor lesion/carcinoma in situ) and potentially impact of the volume of CP/IDC on the results should also be discussed.
6. The GS7 on line 405 (total Gleason score 7?) should be defined.
7. I don’t have the expertise in statistics. Review by statistician may be needed if the editor thinks it deems necessary.
